# Low Temperature Infrared Study of Carbon Monoxide Adsorption on Rh/CeO$_2$

**Hicham Idriss [1,2,*]** and **Jordi Llorca [3]**

[1] Catalysis Department, SABIC R&D at KAUST, Thuwal 23955, Saudi Arabia
[2] Department of Chemistry, University College London, London WC1H 0AH, UK
[3] Department of Chemical Engineering, Institute of Energy Technologies, and Barcelona Research Center in Multiscale Science and Engineering, Technical University of Catalonia, EEBE, 08930 Barcelona, Spain
[*] Correspondence: idrissh@sabic.com or h.idriss@ucl.ac.uk

**Abstract:** Fundamental studies of the interaction of adsorbates with metal oxides alone and on which a noble metal is deposited provide information needed for catalytic reactions. Rh/CeO$_2$ is one of the textbook catalysts for many reactions including syngas conversion to ethanol, water gas shift reaction (WGSR), and ethanol steam reforming. In this work, the adsorption of CO is studied by infrared (IR) spectroscopy, over CeO$_2$ and 0.6 at. % Rh/CeO$_2$ at a temperature range of 90 to 300 K. CeO$_2$ is in the form of nanoparticles with sizes between 5 and 10 nm and exposing predominantly {111} surface termination in addition to non-negligible fraction of the {100} termination, determined from high resolution transmission electron microscopy (HRTEM). The as prepared Rh/CeO$_2$ contained metallic Rh as well Rh cations in higher oxidation states. At 90 K two IR bands were observed at 2183–2186 and 2161–2163 cm$^{-1}$, with the former saturating first. The 2163 cm$^{-1}$ peak was more sensitive to CO pressure than the 2186 cm$^{-1}$. Heating resulted in the depopulation of the 2163 cm$^{-1}$ before the 2186 cm$^{-1}$ peak. The desorption energy computed, assuming a first-order desorption kinetic, was found to be 0.35 eV for the 2186 cm$^{-1}$ and 0.30 for the 2163 cm$^{-1}$ IR peak (+/−0.05 eV). The equilibrium constant at 90 K was computed equal to 1.83 and 1.33 Torr$^{-1}$ for the 2183 and 2161 cm$^{-1}$, respectively. CO adsorption at 90 K on Rh/CeO$_2$ resulted (in addition to the bands on CeO$_2$) in the appearance of a broad band in the 2110–2130 cm$^{-1}$ region that contained two components at 2116 and 2126 cm$^{-1}$. The high frequency of this species is most likely due to adsorption on Rh clusters with very small sizes. The desorption energy of this species was found to be equal to 0.55 eV (+/−0.05 eV). Heating the CO covered Rh/CeO$_2$ surface accelerated the disappearance of CO species over CeO$_2$ and resulted in the appearance of CO$_2$ bands (at about 150 K) followed by carbonate species. At 300 K, the surface was mainly composed of carbonates.

**Keywords:** CO adsorption; Rh/CeO$_2$ nanoparticle; desorption energy; adsorption isotherm; gem-dicarbonyl species; CeO$_2$ (111); CeO$_2$ (100); high resolution transmission electron microscopy (HRETM) of CeO$_2$

## 1. Introduction

The interaction of CO with the surfaces of metal oxides and metal deposited on the metal oxides gives considerable information related to bonding, metal dispersion, and redox properties, among others [1–4]. The weak interaction of CO on oxides in general dictates investigations at low temperature where electrostatic interactions prevail [5]. In that regard, CO adsorption on reducible metal oxides such as CeO$_2$ has been extensively studied experimentally and by computational methods [6–8]. A large fraction of CO interaction with metal oxides at low temperatures is conducted

by infrared spectroscopy [9,10], although most of this work is on stoichiometric and reduced pure metal oxides, and less devoted to metal/metal oxides.

CO interaction with $CeO_2$ model surfaces is studied in good details, and a large body of information is given in a recent review article by Wang and Wöll on $CeO_2$ single crystals and nanorods [11]. In brief, several blue shifted (with respect to gas phase $\nu CO$ at 2146 cm$^{-1}$) IR bands are seen extending from 2178 to 2150 cm$^{-1}$ depending on the coordination number (mostly dictated by crystallographic direction) and the presence of $Ce^{3+}$ cations. Many Density Functional Theory (DFT) computation studies have been conducted to probe into the vibrational frequency modes and adsorption energies of different systems of CO on $CeO_2$ surfaces, among other parameters [12–14]. Ab-initio computation of $CeO_2$ requires the use of either a hybrid functional (DFT-HF) or the incorporation of the Hubbard parameter (U) because of the strong electron exchange correlation making the assignment, in particular for the reduced surfaces less certain [15,16]. Still there is a good agreement between theory and experiments on a few main aspects: (i) That CO vibrational frequency is further blue shifted with respect to gas phase CO on $CeO_2$, (ii) that this vibrational frequency is further blue shifted on a reduced surface, when compared to that on a stoichiometric surface, (iii) that the adsorption energy is slightly higher on reduced $CeO_2$ when compared to the stoichiometric one; while the adsorption energy changes with surface structure, the exact adsorption sites on reduced surfaces is still subject to further studies.

Much less work was conducted for CO adsorption on Rh/$CeO_2$ at a low temperature because the bulk of the work, mostly by IR, was conducted for the purpose of water gas shift reaction (WGSR), syngas conversion, ethanol synthesis from syn gas, and CO oxidation among other reactions [17–20]. The Rh/$CeO_2$ catalysts themselves have been well studied for decades where metallic Rh particles are formed upon reduction, although the as prepared Rh/$CeO_2$ contains Rh particles with several oxidation states and possibilities of Rh ions substitution of Ce cations [21,22]. We have previously observed that at 90 K, CO adsorption on Rh/$CeO_2$ resulted, in addition to the two IR bands attributed to CO adsorbed on $CeO_2$, in the appearance of a broad IR band at about 2120 cm$^{-1}$ attributed to adsorbed CO on small particles of Rh [23], based on similarly observed frequencies of Rh clusters over $Al_2O_3$ [24]. Others have also found, on Rh deposited on $CeO_2$ thin film, a band with a $\nu CO$ frequency at 2100 cm$^{-1}$ and attributed it to the symmetric stretch of a gem-dicarbonyl species [17] on small Rh-O particles by Reflection Absorption Infra Red Spectroscopy (RAIRS). The asymmetric stretch expected to be at about 2030 cm$^{-1}$ was not observed due to selection rules. As presented in this work, it was possible to see two peaks on Rh/$CeO_2$ upon CO adsorption at a high coverage, in this region. These were at 2126 and 2114 cm$^{-1}$ with no evidence of bands due to an asymmetric stretch indicating that these species are most likely due to linearly adsorbed CO on small clusters of Rh cations and not due to a gem dicarbonyl species.

In this work we focus on the irreversible adsorption of CO over $CeO_2$ and Rh/$CeO_2$ at 90 K as a function of coverage, followed by the reversible adsorption as a function of CO pressure, then on the effect of temperature from which we extract the desorption energy. We also observed some transformation of adsorbed CO into carbonate species upon heating to temperatures above 120 K.

## 2. Results and Discussion

### 2.1. CO Adsorption on CeO₂

#### 2.1.1. Irreversible Adsorption

Infrared spectra of sequential irreversible CO adsorption on $CeO_2$ at 90 K followed by evacuation (10$^{-4}$ Torr) are presented in Figure 1A. The introduction of small amounts of CO (0.27 Torr) leads to the appearance of a band in the $\nu$C–O stretching region with a maximum at 2186 cm$^{-1}$. A gradual increasing of the CO pressure (0.40 Torr) resulted in its intensity increase and an emergence of a lower wavenumber shoulder at about 2165 cm$^{-1}$. Further increase of the CO pressure enhanced the intensity of both bands. The shoulder at 2165 cm$^{-1}$ eventually shared similar absorbance intensity as the band at 2186 cm$^{-1}$ at high coverages. At saturation coverage, both absorption bands experienced slight

red shift to 2182 and 2163 cm$^{-1}$, respectively. Although the range of these shifts are within the experimental error, this type of shift is commonly observed for sequential CO adsorption at high surface coverage [25–27]. The assignment of these two bands is not definitive. Two C–O stretching bands of 2170 and 2150 cm$^{-1}$ [28], or 2168 and 2157 cm$^{-1}$ were also observed by others [29]. These two bands are assigned to CO adsorbed on Ce$^{4+}$ cations and defects associated with Ce$^{3+}$, respectively. While XPS Ce3d of CeO$_2$ shows the common complete oxidation, small amounts of surface Ce$^{3+}$ may not be captured (due to rising background of Ce3d lines). Yet, the similar contributions of both peaks indicate that they most likely originate from different states with similar fractions. In other words, it is not possible that trace amount of Ce$^{3+}$ (undetected by XPS) would give rise to a CO band of equal intensity to that observed on Ce$^{4+}$ cations. These are most likely associated with different crystallographic planes rather than different oxidation states. Figure 1B,C shows two representative high resolution transmission electron microscopy (HRTEM) images of the CeO$_2$ used in the study. It is clear that they represent lattice fringes mostly with {111} and {100} directions and in a few cases (not shown) lattice spacing related to {110}-termination were present. It is worth mentioning that the (100) surface of the fluorite structure is unstable and reconstructs to other facets [30–32]. By analogy to the single crystal studies[11], where the fully oxidized CeO$_2$(111) and CeO$_2$(110) crystals gave bands at 2154 and 2175 cm$^{-1}$ and the fact that on CeO$_2$ nanorods [8], these two bands were observed with similar intensities, the two bands observed in this work may be assigned to adsorption on Ce$^{4+}$ cations on two different surface terminations. The higher frequency observed here might be linked to crystallite size where unlike single crystals, nano-rods, or large particles (above 30 nm or so), the presence of small particles can induce lattice expansion that may in turn affect the vibrational frequency of an adsorbate.

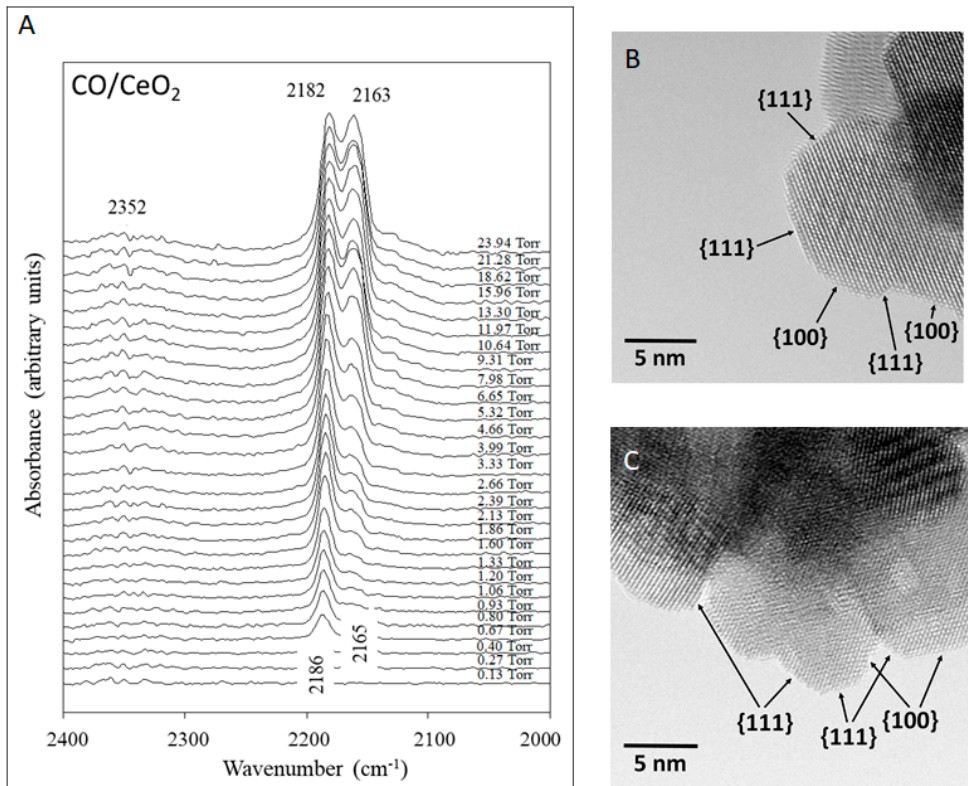

**Figure 1.** (**A**) Fourier Transform Infra Red (FT-IR) spectra of CO adsorbed on CeO$_2$ at 90 K. Indicated pressures are those of CO (upon exposure at the indicated pressure for 1 min each), spectra are collected at ~10$^{-4}$ Torr. (**B**,**C**) High resolution transmission electron microscopy (HRTEM) of the CeO$_2$ particles on which the surface termination is indicated.

Figure 2A illustrates the dependence of the absorption intensity of the bands at 2186 and 2165 cm$^{-1}$ on CO exposure. For both peaks, up to 2 Torr, the variation of this irreversible adsorption is linear.

The changing in the shape of the curve above 4 Torr for the 2186 cm$^{-1}$ peak and above 8 Torr for the 2163 cm$^{-1}$ peak indicate surface saturation (Figure 2B,C will be commented after Figure 3).

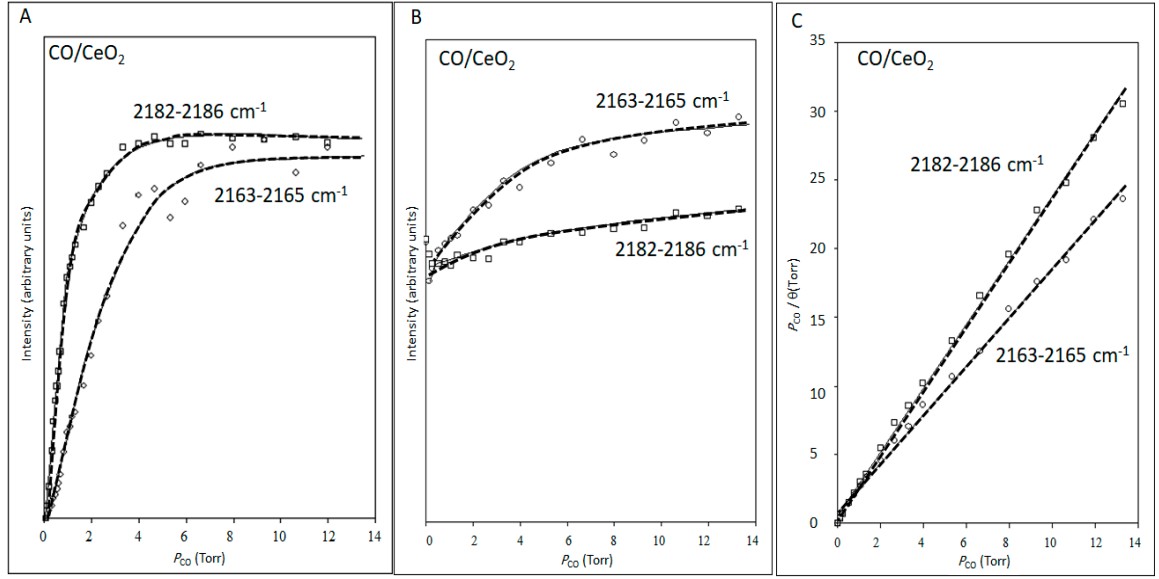

**Figure 2.** (**A**) plot of dependence of the adsorbed intensity of the CO bands (□) 2186 cm$^{-1}$ and (○) 2165 cm$^{-1}$ at 90 K from (A). (**B**) A plot of absorption bands intensity ((□) 2186 cm$^{-1}$ and (○) 2163 cm$^{-1}$) versus CO adsorption pressure on CeO$_2$ at 90 K. (**C**) Adsorption isotherm upon CO adsorption on CeO$_2$ (□) 2186 cm$^{-1}$ and (○) 2163 cm$^{-1}$ at 90 K.

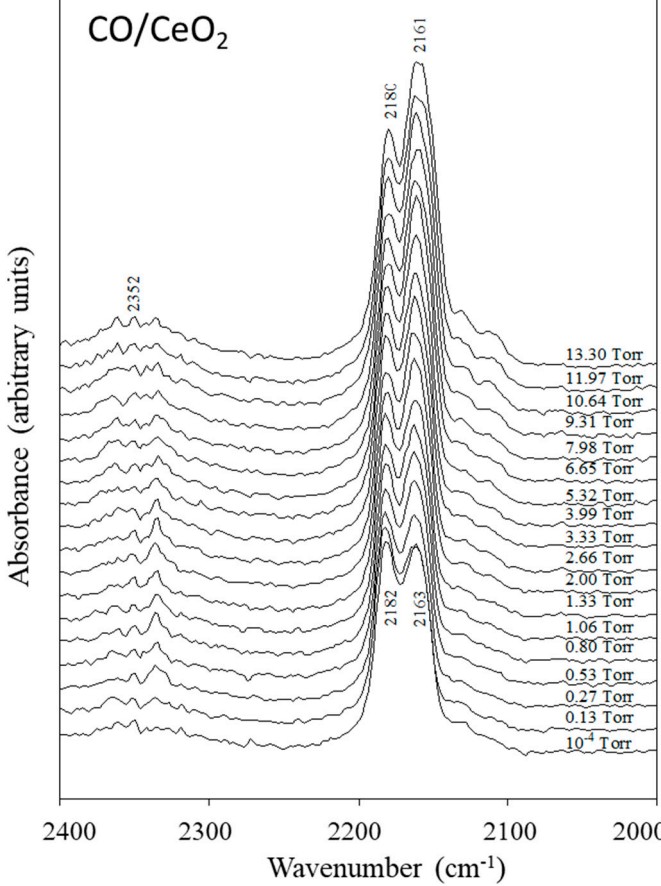

**Figure 3.** FT-IR spectra of CO adsorbed on CeO$_2$ at 90 K; indicated pressures are real working pressures.

### 2.1.2. Reversible Adsorption

Figure 3 shows sequences of the carbonyl infrared spectra obtained following further adsorption of CO on $CeO_2$ at 90 K under reversible adsorption conditions. Upon increasing the surface coverage, small increases were observed for the band at 2182 $cm^{-1}$, while that at 2163 $cm^{-1}$ showed gradual increase in intensity. After 0.53 Torr of CO dosing, the band at 2163 $cm^{-1}$ predominates over the band at 2182 $cm^{-1}$. The increases experienced by both bands were reduced to their initial height immediately after evacuation. If the two bands are attributed to adsorption on two different surfaces, then one may conclude that the one still responding to higher pressures has a weaker surface interaction. Figure 2B shows a plot of absorption band intensity versus CO pressure, while Figure 2C gives a plot of the adsorption isotherm of CO on $CeO_2$ at 90 K (the intensity of the two bands in Figure 3). The adsorption coefficients can be calculated from the intercept of the adsorption isotherm plot. They were found to be equal to 1.83 and 1.33 $Torr^{-1}$ for the 2183 and 2161 $cm^{-1}$ bands, respectively.

### 2.1.3. Temperature Effect

Figure 4A shows the effect of temperature on the irreversibly adsorbed CO on $CeO_2$. Clearly both bands do not decrease together. The band at 2163 $cm^{-1}$ is far more sensitive to increasing temperatures than the 2182 $cm^{-1}$ band. This further confirms that both bands do not originate from the same surface sites. It is worth noticing the band corresponding to $CO_2$ at 2350 $cm^{-1}$, indicating that part of the adsorbed CO is oxidized to $CO_2$ at these low temperatures. $CO_2$ is equally weakly adsorbed on $CeO_2$ and as such its detection does not successively translate the amount formed in part due to the inevitable back ground contribution. At 210 K all CO as well as $CO_2$ is desorbed. Both bands at 2182 and 2163 $cm^{-1}$ underwent a high wavenumber shift, back to their original values at low coverage at 90 K, to 2186 and 2165 $cm^{-1}$ before their disappearance. From both, the effect of temperature and pressure, one observes that the band at 2163 $cm^{-1}$ is more sensitive to change in conditions when compared to the band at 2186 $cm^{-1}$; increasing the pressure increases its intensity and increasing the temperature decreases its population on the surface (weaker bonding energy). Taking the peak area of νCO at 90 K as a representative of full coverage and plotting the changes as a function of temperature we can extract the desorption energy. Since the desorption is expected to be first-order, we may take a pre-factor of $10^{13}$ $s^{-1}$ and the desorption energy can be computed at $\Theta = 0.5$ (from the corresponding temperatures as indicated by the arrows on Figure 4B). The desorption energies for the bands at 2182 and 2163 $cm^{-1}$ were found to be equal to 36 and 32 kJ/mol $\pm$ 5 kJ/mol, respectively (0.35 and 0.3 eV $\pm$ 0.05 eV, respectively). This is close to those reported by others[11] on single crystal experimentally (0.27 and 0.31 eV from $CeO_2$ (111) single crystal) and those computed using the DFT + U method (0.4 eV).

### 2.2. CO Adsorption on $Rh/CeO_2$

#### 2.2.1. Irreversible Adsorption

Infrared spectra of sequential irreversible CO adsorption on $Rh/CeO_2$ at 90 K and after evacuation ($10^{-4}$ Torr) are presented in Figure 5. The initial CO adsorption (0.13 Torr) resulted in the appearance of a broad band located at ca. 2124 $cm^{-1}$. This band showed gradual increase in intensity with increasing in CO coverage (at saturation it is composed of two bands at 2126 and 2114 $cm^{-1}$). Increasing the CO pressure to 0.40 Torr resulted in the emergence of a second band at 2190 $cm^{-1}$. This band also increased in intensity with increasing CO pressures. Meanwhile, a small band was observed at ca. 2350 $cm^{-1}$ and is due to the formation of $CO_2$ on the surface. At 1.06 Torr of CO, a shoulder at the 2190 $cm^{-1}$ band was evolved and positioned at 2178 $cm^{-1}$. This third band steadily increased in intensity with increasing CO pressures. At saturation, a broad shoulder at about 2030 $cm^{-1}$ was clearly visible (see Figure 8B spectrum a for a clear display). The bands at 2124 and 2030 $cm^{-1}$ (broad) were not observed on $CeO_2$ and are thus attributed to the presence of Rh. Other works have shown that CO adsorption on $Rh/Al_2O_3$ at 120 K gave two adsorption bands featured at 2104 and 2038 $cm^{-1}$,

which were attributed to the symmetric and asymmetric stretches of the Rh bound gem-dicarbonyl $Rh^+(CO)_2$ species, respectively [33]. Two other bands located at 2069 and 1923 $cm^{-1}$ were also detected and are assigned to CO adsorbed on linearly $Rh^+(CO)$ and bridging $Rh_2(CO)$ modes [34]. However, works done on CO adsorption on oxidized $Rh/Al_2O_3$ catalysts saw the appearance of a band located at 2125 $cm^{-1}$, and was assigned to CO adsorbed on $Rh^{2+}$ sites [35]. By analogy, the band at 2124 $cm^{-1}$ is attributed to $Rh^{2+}(CO)$ surface species, while the weak broad band centered at ca. 2030 $cm^{-1}$ is most likely due to multiple species, $Rh^+(CO)$ as well as $Rh^+(CO)_2$ species. Figure 6 illustrates the intensity dependence of the $\nu(CO)$ bands (2190, 2178, and 2124 $cm^{-1}$) on the amount of CO introduced into the cell. All three bands reached saturation at ca. 10.6 Torr min. of CO. Also noted is a decrease in the intensity at ca. 1.2 Torr min, although further increasing in CO coverage recovered the intensity of these bands. A given IR signal intensity of an adsorbate is a function of its coverage. Yet because the vibrational mode of a functional group is affected by its dipole moment, changes in the order of a surface coverage can change the dipole moment which in turn may non-linearity affect its signal. This is often manifested in the case of CO where changes in surface coverage collectively change the dipole moment resulting in intensity variations of $\nu CO$ [36–38].

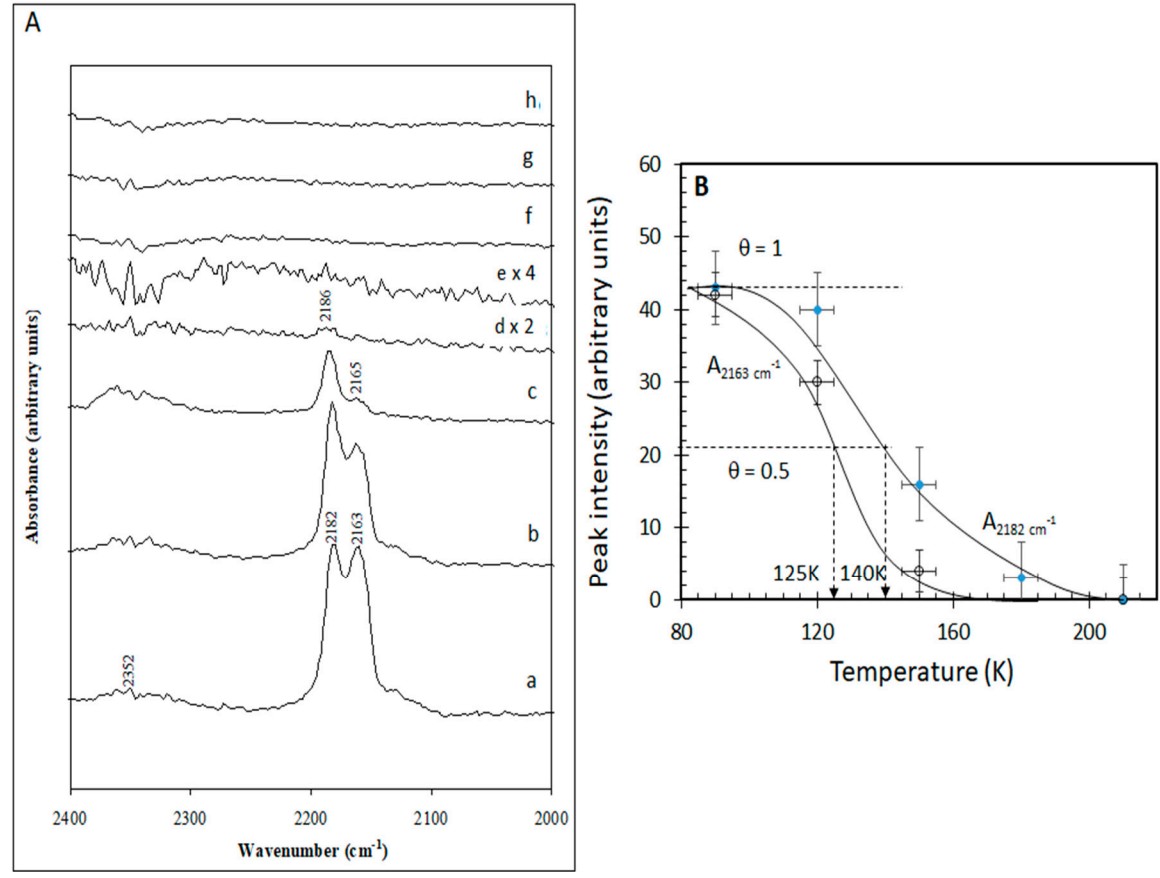

**Figure 4.** (**A**) FT-IR spectra of CO adsorbed on $CeO_2$ at (a) 90 K, (b) 120 K, (c) 150 K, (d) 180 K, (e) 210 K, (f) 240 K, (g) 270 K, and (h) 290 K. (**B**) Change of coverage with temperature, the surface coverage, $\nu$, at 90 K was taken as 1.

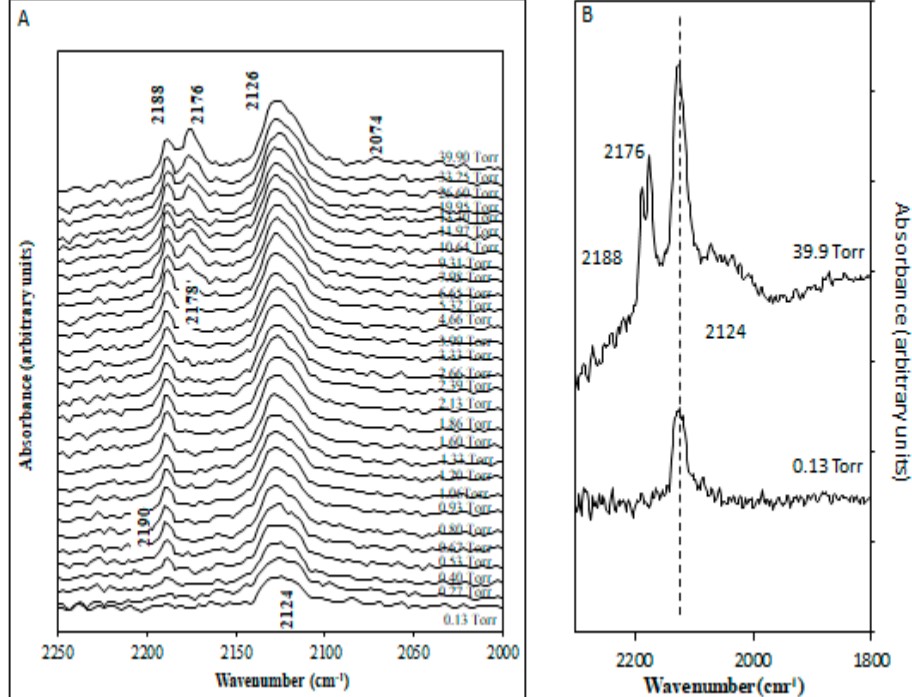

**Figure 5.** (**A**) FT-IR spectra of CO adsorbed on Rh/CeO$_2$ at 90 K. Indicated pressure are those of CO (upon exposure at the indicated pressure for 1 min each), the working pressure is ~10$^{-4}$ Torr. (**B**) Spectra obtained at the first (0.13 Torr) and at the last (39.9 torr) CO exposure.

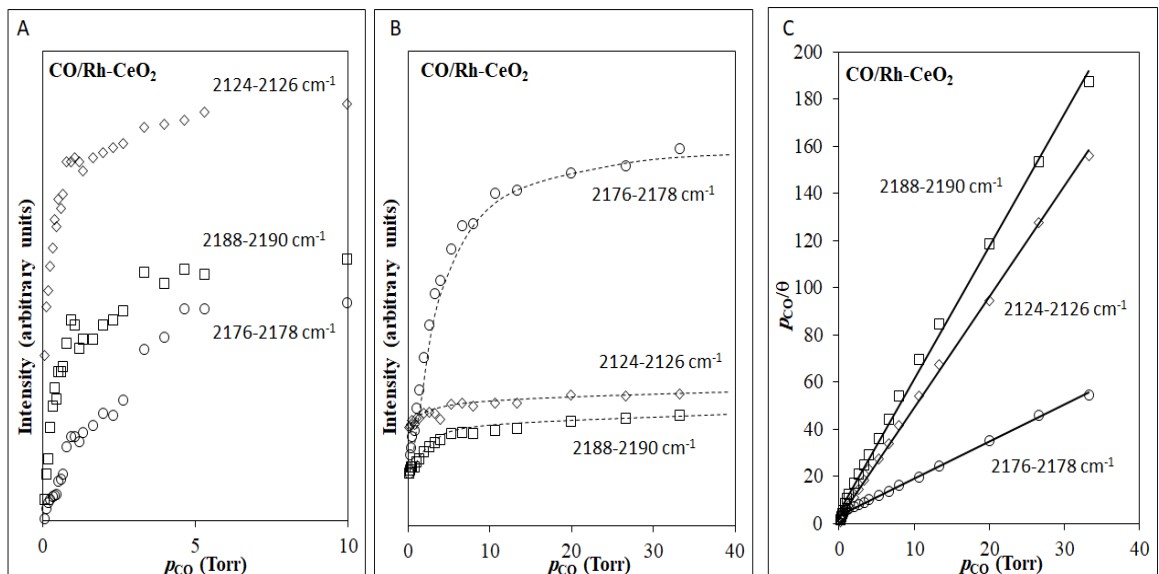

**Figure 6.** (**A**) A plot of dependence of the adsorbed intensity of the CO bands (□) 2190 cm$^{-1}$, (○) 2178 cm$^{-1}$, and (◇) 2124 cm$^{-1}$ at 90 K. (**B**) A plot of absorption bands intensity ((□) 2190 cm$^{-1}$, (○) 2178 cm$^{-1}$, and (◇) 2124 cm$^{-1}$) versus CO adsorption pressure on Rh/CeO$_2$ at 90 K. (**C**) Adsorption isotherm of CO adsorption on Rh/CeO$_2$ (□) 2190 cm$^{-1}$, (○) 2178 cm$^{-1}$, and (◇) 2124 cm$^{-1}$ at 90 K.

### 2.2.2. Reversible Adsorption

Figure 7 shows the infrared spectra obtained from sequences of CO adsorption on Rh/CeO$_2$ at 90 K under reversible adsorption conditions. It is evident that the band at 2176 cm$^{-1}$ showed significant increases in intensity and shifted with increasing coverage to a lower wavenumber up to 2169 cm$^{-1}$, while bands at 2188, 2124, and 2074 cm$^{-1}$ showed small variations in intensity and negligible frequency

shifts with increasing coverage. All three bands dropped back to their initial height immediately after evacuation. The considerable shift in frequency of the band at 2176 cm$^{-1}$ (not observed on CeO$_2$) and increase in intensity might be linked to the presence of Rh ions on CeO$_2$. If we assume that the νCO frequency at 2176 cm$^{-1}$ is due to adsorption on the reconstructed (100) surface, then it might be possible that Rh ions are preferentially adsorbed on these planes; the high surface energy when compared to the (111) surface [39] may result in a more favorable interaction with Rh ions during the impregnation or calcination steps.

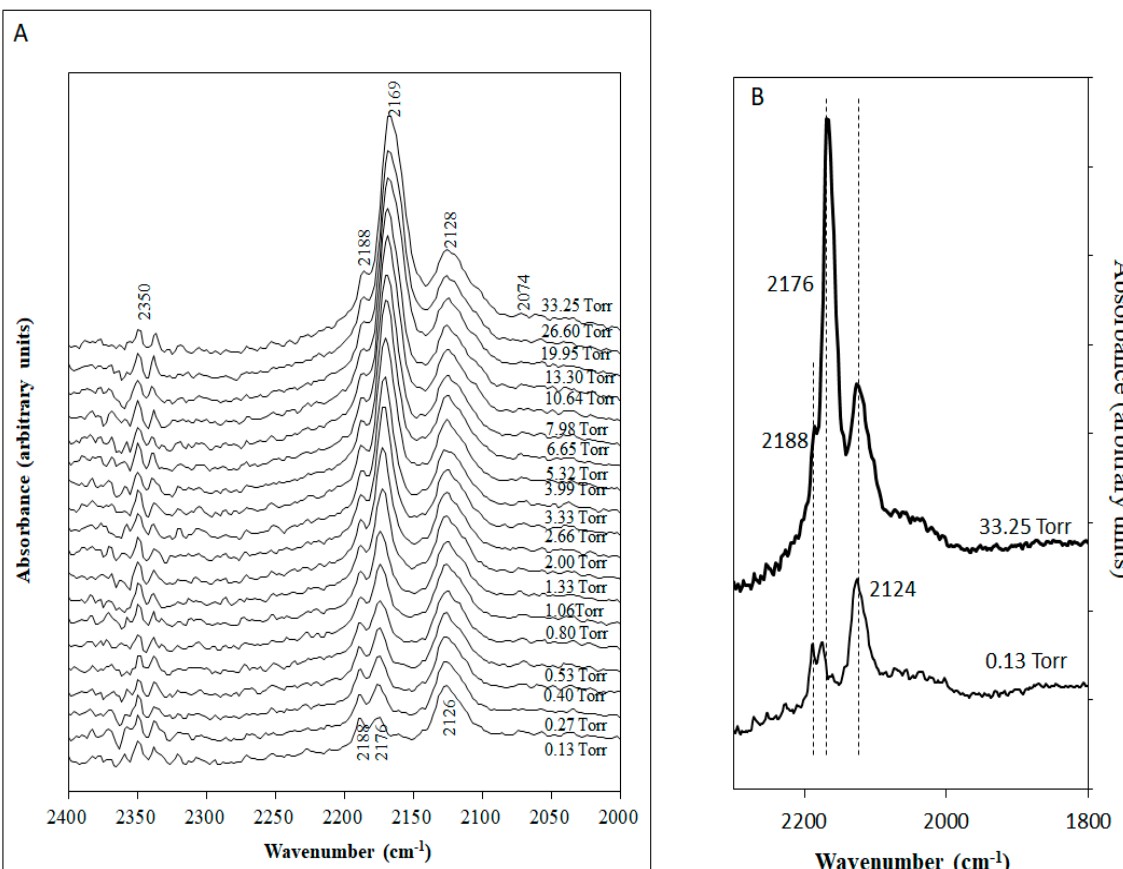

**Figure 7.** (**A**) FT-IR spectra of CO adsorbed on Rh/CeO$_2$ at 90 K. Indicated pressure is real working pressure. (**B**) Spectra obtained at the first (0.13 Torr) and at the last (33.25 Torr) CO exposure.

Figure 6B shows a plot of absorption band intensity versus CO pressures, and Figure 6C gives a plot of the adsorption isotherm of CO adsorbed on Rh/CeO$_2$ at 90 K. The adsorption coefficients obtained are 0.20, 0.35, and 0.53 Torr$^{-1}$ for the bands located at 2188, 2178, and 2124 cm$^{-1}$, respectively.

### 2.2.3. Temperature Effect

Figure 8A,B shows the carbonyl infrared spectra of the temperature dependence of CO adsorbed on Rh/CeO$_2$ at 90 K. By 120 K immediate reduction of the band at 2188 cm$^{-1}$, and disappearance of the band at 2176 cm$^{-1}$ is observed. The band at 2124 cm$^{-1}$ also showed a slight reduction in intensity. At 150 K, the band at 2188 cm$^{-1}$ was vanished, while broad bands at ca. 2350 and in the 2000–2100 cm$^{-1}$ showed a considerable growth in their intensities. They are contributed to the adsorbed CO$_2$ and linearly CO adsorbed on Rh clusters and most likely gem-dicarbonyl species (the convolution of the peaks makes it difficult to have a clear assignment), respectively. Further increase in temperature saw gradual decrease in the intensity of the bands at ca. 2350 and 2124 cm$^{-1}$. Increasing the temperature to 210 K results in vanishing of the band at ca. 2350 cm$^{-1}$, and a gradual decrease of the bands at 2124 and ca. 2030 cm$^{-1}$. Both bands were still visible when 290 K was reached. The emergence of

bands in the carbonate region [40] is clear with increasing temperatures (1597 and 1682 $cm^{-1}$), it is thus possible that CO adsorbed on Rh has resulted in its oxidation to $CO_2$ which further interacted with surface oxygen making carbonate species. CO adsorption as CO or as carbonates has been studied by others on $CeO_2$. It was found that the adsorption energy is far higher when CO is adsorbed as carbonates [41]. On $Ce_{0.875}Fe_{0.125}O_2$ the reaction was studied by DFT + U and found to occur without an activation energy [42]. Figure 8C presents the changes in the peak intensity at 2126 $cm^{-1}$ with temperature. There is a monotonic decay with a slope change at about 200 K, most likely due to changes in surface coverage, affecting the dipole moment, although it can be due to faster channel for surface transformation as it coincides with the increase in the carbonate signal. From the coverage at 0.5 and taking a frequency factor equal to $10^{13}$ $s^{-1}$, assuming a first-order desorption, the desorption energy is computed to be equal to 0.55 +/− 5 eV (or about 53 kJ/mol). The overall frequency trend with coverage and temperature is similar to what has been reported previously for $Rh/CeO_2$ thin film by IRAS where the authors found at 120 K a $\nu$CO band at 2100 $cm^{-1}$, that decreased with temperature, attributed to gem dicarbonyl species of Rh oxide particles with very small size. While this explanation is possible it may also be related to CO adsorbed at the interface $Rh$-$CeO_2$ because its disappearance coincides with peaks associated to adsorbed $CO_2$ and carbonates.

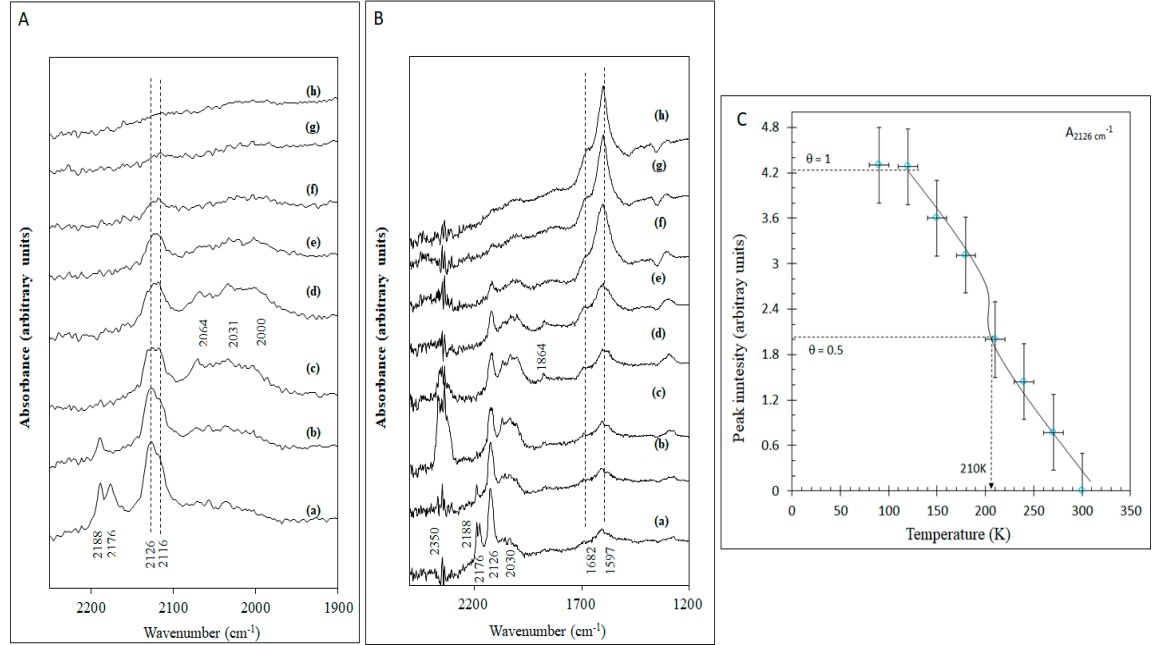

**Figure 8.** (**A**) FT-IR spectra of CO adsorbed on $Rh/CeO_2$ in the 2250–1900 $cm^{-1}$ region at (a) 90 K, (b) 120 K, (**c**) 150 K, (d) 180 K, (e) 210 K, (**f**) 240 K, (g) 270 K, and (h) 290 K. (**B**) A FT-IR spectra of CO adsorbed on $Rh/CeO_2$ in the 2500–1200 $cm^{-1}$ region at (a) 90 K, (b) 120 K, (c) 150 K, (d) 180 K, (e) 210 K, (f) 240 K, (g) 270 K, and (h) 290 K. (**C**) Peak intensity of the $\nu$CO at 2126 $cm^{-1}$ as a function to temperature, the coverage, $\nu$, is taken as equal to 1 at 90 K.

## 3. Experimental

Cerium oxide ($CeO_2$) was prepared by precipitating cerous nitrate ($Ce(NO_3)_3 \cdot 6H_2O$) (100 g/dissolved in 0.4 L de-ionized water while stirring at 373 K with ammonia (0.91 $molL^{-1}$) at pH of 8). The resulting white slurry was collected by filtration, washed with de-ionized water, and dried at 373 K for 12 h. The dried powder was calcined in a furnace at 773 K for 4 h in air. The sample was then ground with a mortar and pestle. $Rh/CeO_2$ was prepared by the conventional impregnating technique, where the $CeO_2$ support and appropriate amounts of Rh cations (from a stock solution of 0.4 g $L^{-1}$ of $Rh^{3+}$ cations (ex $RhCl_3 \cdot xH_2O$; amount of $Rh^{3+}$ is 38%, Sigma Aldrich) in 1 M HCl) were mixed in a beaker at ambient temperature with continuous stirring. The temperature was than raised

to 373 K and retained to boil off the liquid. When most of the liquid vaporized, forming a paste like material, it was further dried in an oven for 12 h at 373 K. The dried powder was then calcined in a furnace for 4 h at 673 K under flowing air. The BET surface area of $CeO_2$ was found = 33 $m^2/g_{Catal.}$ with marginal changes upon Rh deposition. XRD showed the characteristic fluorite structure of $CeO_2$ with no evidence of Rh crystallites due to their small concentration (1 wt. %). The particle size of $CeO_2$ is 10–15 nm from XRD and TEM measurements [43]. High-resolution transmission electron microscopy (HRTEM) was carried out at 200 kV with a JEOL JEM 2100 instrument equipped with a $LaB_6$ source (Musashino, Akishima, Tokyo, JAPAN). The point-to-point resolution of the microscope was 0.19 nm. Samples were deposited on holey-carbon-coated Cu grids from alcohol suspensions. XPS Rh3d showed a doublet at 308.7 and 312.3 eV attributed to Rh $3d_{5/2}$ and Rh $3d_{3/2}$ lines, respectively. The broad FWHM of 3.0 eV of Rh $3d_{5/2}$ may suggest the presence of more than one species. Rh was found to be at ca. 0.6 at % with a Rh/Ce ratio of 0.014 (theoretical ratio is 0.017).

Infrared spectra were collected using a Digilab FTS-60 Fourier transform spectrometer (Hopkinton, MA, USA), with a 256-scan data acquisition at a resolution of 4 $cm^{-1}$. CO adsorption was performed in a stainless steel IR cell equipped with removable $CaF_2$ windows (32 mm diameter, 4 mm thick) sealed with Viton O-rings. The $CaF_2$ windows allowed monitoring of the 4500 to 1000 $cm^{-1}$ region and the cell operated at temperatures between 77 and 773 K. The catalysts were pressed into a self-supporting disc (ca. 15 mm in diameter) and mounted onto a gold-plated brass sample holder. The IR cell was connected to a vacuum line and maintained at a base pressure (ca. $10^{-5}$ Torr) with a roughing pump backed up by a diffusion pump (Figure 9). $O_2$ and CO were contained in separate glass bulbs, attached to the vacuum line. Catalysts were annealed under 20 Torr of oxygen at 673 K followed by evacuation for 1-h prior to conducting the experiments.

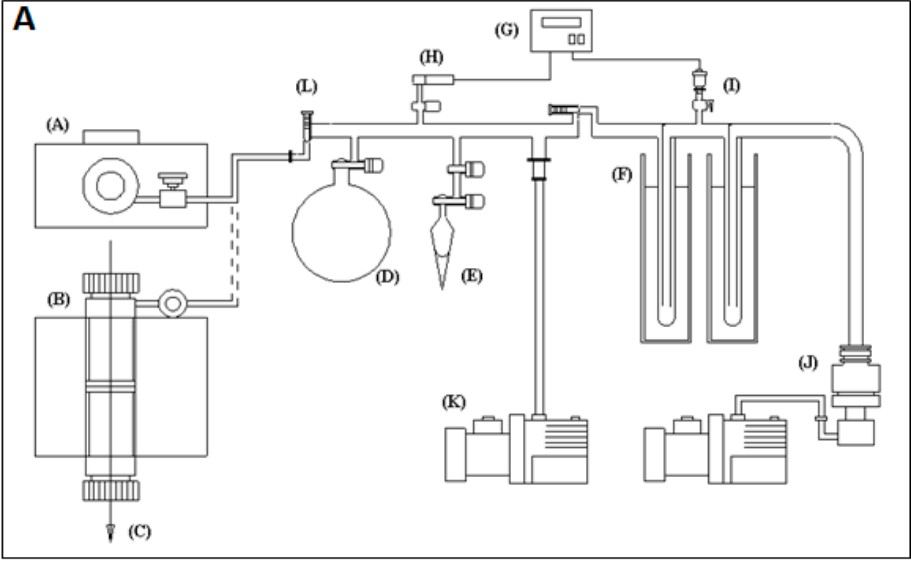

**Figure 9.** *Cont.*

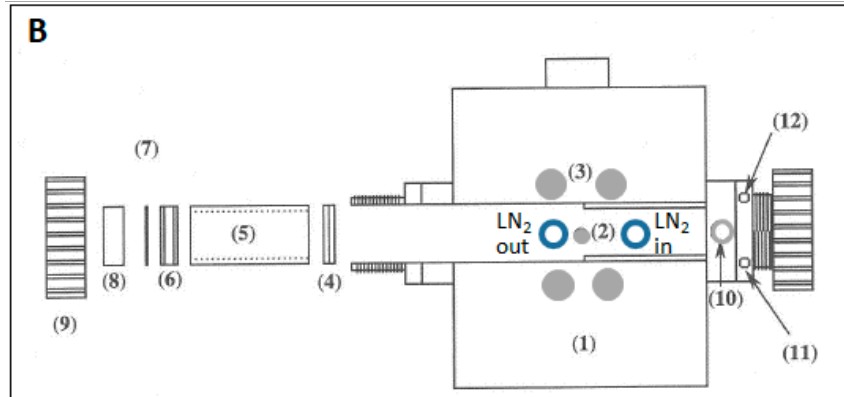

**Figure 9.** (**A**) Schematic diagram of the vacuum system and infrared (IR) cell used; (A) side view of IR cell, (B) top view, (C) IR beam, (D) gas bulb, (E) liquid bulb, (F) cold trap, (G) pressure gauge, (H) pirani probe head, (I) penning probe head, (J) diffusion pump, (K) roughing pump, and (L) Teflon valves. (**B**) A semi cross-sectional view of the in situ infrared cell. (1) Stainless steel body, (2) thermocouple, (3) cartridge heaters, (4) catalyst sample holder, (5) spacer, (6) $CaF_2$ windows, (7) Viton O-ring, (8) Teflon spacer, (9) screw cap filling, (10) connection to vacuum system, (11) cooling inlet, (12) cooling outlet.

## 4. Conclusions

The adsorption of CO is studied by IR spectroscopy over nanoparticles of $CeO_2$ (in the range 5–10 nm in size) at 90 K and resulted into the formation to two bands at 2183–2186 and 2161–2163 $cm^{-1}$, with the former saturating first. The computed desorption energy, assuming a first-order desorption kinetic was found to be 0.35 eV for the 2186 $cm^{-1}$ and 0.30 for the 2163 $cm^{-1}$ IR peak (+/−0.05 eV). CO adsorption at 90 K on $Rh/CeO_2$ resulted (in addition to the bands on $CeO_2$) in the appearance of a broad band in the 2110–2130 $cm^{-1}$ region, that contained two components at 2116 and 2126 $cm^{-1}$ attributed to the adsorption on Rh cations of Rh clusters. This species was more stable than those on $CeO_2$ (it polluted the surface first and had a higher desorption energy, equal to 0.55 eV (+/−0.05 eV)). Heating the CO-covered $Rh/CeO_2$ surface accelerated the disappearance of CO species from $CeO_2$ at 150 K, and only species related to CO adsorption on Rh centers were found. The disappearance of CO on Rh particles coincided with the formation of carbonate species which dominated the surface at 300 K.

**Author Contributions:** Conceptualization (H.I.), Methodology (H.I.), Investigation (H.I. and J.L.), Writing-Original Draft (H.I. and J.L.).

**Funding:** This research received no external funding.

**Acknowledgments:** The work is part of the MSc thesis of Po Yo sheng now at Advanced Optoelectronic Technology, Inc., Information Technology and Services, Hsinchu Industrial Park, Taiwan.

**Conflicts of Interest:** The authors declare no conflict of interest.

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
