# Peer review of "Low Temperature Infrared Study of Carbon Monoxide Adsorption on Rh/CeO2"

_catalysts, doi:10.3390/catal9070598_

Round 1

Reviewer 1 Report

The work is novel and provides great insight into the field. I recommend this paper for publication. Just mark a and b on the figures.

Author Response

The work is novel and provides great insight into the field. I recommend this paper for publication. Just mark a and b on the figures.  Thank you, OK

Reviewer 2 Report

The authors report a low temperature IR study of CO adsorption on Rh/CeO2. The “irreversible” CO adsorption was already reported by the authors in Ref. 23; now the adsorption under “real” working pressures was additionally reported. This differentiation is a little bit confusing. It is clear that the amount of adsorbed CO depends on the partial pressure of CO and the strength of the interaction of CO with the respective metal site. For detecting all metal sites an optimal partial pressure of CO is needed as can be determined experimentally. This is obviously the case at ca. 13 Torr. Therefore, to make it more clearly for the reader which Ce and Rh sites are present in the sample, a comparative Figure is more helpful than the presented Figures 1, 3, 5, and 7. Here, the spectra of CeO2 and Rh/CeO2 at 13 Torr CO before and after evacuation should be shown. The pressure dependence of band intensities can be shown separately (cf. Figure 2 and 6). But the Figure captions have to be revised, because they are unclear in the presented form. Figure 4B is confusing and can be omitted.

By inspecting the broad Rh-CO band it seems that there are more components present, not only the two geminal Rh(I) dicarbonyl bands. Normally, these bands appear clearly separated in a distance of ca. 70 cm-1. Are the authors sure that only Rh(I) is present; or what is the oxidation state of Rh in the “Rh-oxide” nanoparticles? Considering that the catalysts were pretreated oxidatively, the assignment has to be revisited.

Concerning the experimental part: Did the IR really cell operated in the whole temperature range 673-90K ? How CO was dosed and the pressure was measured? Here, more detailed information also of the cell design would be helpful.

Line 184: Where is the broad shoulder about 2030 cm-1 in Figure 8B?

Line 189: Ref. 128?

Line 194: What “Torr min” means; see also Fig. 2A and 6A?

Line 196/197: The deeper meaning of this sentence is not visible.

Author Response

Referee 2

The authors report a low temperature IR study of CO adsorption on Rh/CeO2. The “irreversible” CO adsorption was already reported by the authors in Ref. 23; now the adsorption under “real” working pressures was additionally reported. This differentiation is a little bit confusing. …

For detecting all metal sites an optimal partial pressure of CO is needed as can be determined experimentally.

For irreversible adsorption, it is the exposure (pressure x time) that matters not the final pressure.

This is obviously the case at ca. 13 Torr. Therefore, to make it more clearly for the reader which Ce and Rh sites are present in the sample, a comparative Figure is more helpful than the presented Figures 1, 3, 5, and 7.  Here, the spectra of CeO2 and Rh/CeO2 at 13 Torr CO before and after evacuation should be shown. …. But the Figure captions have to be revised

We have opted to keep the figures, improved the figure captions, added headings, and added as suggested two comparative figures for CO on Rh/CeO2 (at low and high CO pressures). 

Figure 4B is confusing and can be omitted.

Figure 4B (now Figure 5B) is used to extract the desorption energy, we have explained it further in the text as well as in the figure caption.

By inspecting the broad Rh-CO band it seems that there are more components present, not only the two geminal Rh(I) dicarbonyl bands. Normally, these bands appear clearly separated in a distance of ca. 70 cm-1. Are the authors sure that only Rh(I) is present; or what is the oxidation state of Rh in the “Rh-oxide” nanoparticles?

We are not, we are sure that RhO2 exists (HRTEM and XPS of the same catalyst has been reported a few times in the past).  This is simply due to thermodynamic (the catalysts have been oxidized by O2 prior to experiments, indicated in the experimental section). 

Considering that the catalysts were pretreated oxidative, the assignment has to be revisited.

We have taken the referee’s point and kept only the clear assignment of CO on Rh.  The peak at ca. 2120 cm-1 is clear yet the absence of a similar peak at about 2030-2040 cm-1 (that would be due the asymmetric stretch of a gem di-carbonyl species) favors the attribution to a linearly adsorbed mode.  Based on the high wavenumber (when compared to that observed on Rh metal) and its very weak binding energy (about 0.5 eV) also when compared to that of CO on Rh metal, this peak is due to adsorption on Rh cations.

Concerning the experimental part: Did the IR really cell operated in the whole temperature range 673-90K ?

Yes, as indicated in the experimental section.

How CO was dosed and the pressure was measured?

With pressure gauges (TCD up to 10-2 Torr (roughing pump); cold cathode gauge for pressures below).

Here, more detailed information also of the cell design would be helpful.

We have added a figure of the cell design  and the system used in the experimental section (Figure 1 now).

Line 184: Where is the broad shoulder about 2030 cm-1 in Figure 8B?

Now added on the figure.

Line 189: Ref. 128?

Corrected, thank you.

Line 194: What “Torr min” means; see also Fig. 2A and 6A?

Better clarified.  The exposure is a function of both the pressure and time.  We have removed the “min” and indicated in the figure caption that the exposure was conducted for one minute at each pressure.

Line 196/197: The deeper meaning of this sentence is not visible.

Re-written.  The intensity of an IR signal may change non-linearly due to changes in the dipole moment that results from changes in the mode and order of an adsorbate.  This is particularly manifested in the case of CO adsorption on noble metals.

We have paid more attention to wordings, grammar, style, etc... in the revised version.

Thank you very much for your comments and suggestions.

Hicham

Reviewer 3 Report

By means of IR spectroscopy the authors shed light on the absorption properties of carbon monoxide on the surface of CeO2 and Rh/CeO2 nanoparticles at different temperatures. Given the relevant applications of these metallic systems as catalysts for different transformations in which CO is involved, the present study is of undoubted interest for the broad audience of Catalysts. Acceptance is recommended after addressing the following minor points.

- In the experimental part the authors should indicate the exact nature of the Rh salts employed for the elaboration of the Rh/CeO2 system.

- The abbreviation WGSR employed in the introduction should be defined.

- Line 68: Rh oxide instead of Ru oxide.

Author Response

…. Acceptance is recommended after addressing the following minor points.

- In the experimental part the authors should indicate the exact nature of the Rh salts employed for the elaboration of the Rh/CeO2 system.

Added.

- The abbreviation WGSR employed in the introduction should be defined.

Defined

- Line 68: Rh oxide instead of Ru oxide.

The sentence was removed in the revised version.

Round 2

Reviewer 2 Report

Most of the comments were considered now. The manuscript is recommended for publication.